# Characterization of Flax and Hemp Using Spectrometric Methods

Luisa Andronie [1], Ioana Delia Pop [2], Rodica Sobolu [2], Zoriţa Diaconeasa [3], Alina Truţă [4], Cristina Hegeduş [1,*] and Ancuţa Rotaru [1,*]

1 Department of Fundamental Science, University of Animal Sciences and Veterinary Medicine Cluj-Napoca, 400372 Cluj-Napoca, Romania; luisa.andronie@usamvcluj.ro

2 Department of Land Measurements and Exact Sciences, University of Animal Sciences and Veterinary Medicine Cluj-Napoca, 400372 Cluj-Napoca, Romania; popioana@usamvcluj.ro (I.D.P.); rodica.sobolu@usamvcluj.ro (R.S.)

3 Department of Food Science, University of Animal Sciences and Veterinary Medicine Cluj-Napoca, 400372 Cluj-Napoca, Romania; zorita.diaconeasa@usamvcluj.ro

4 Department of Forestry, University of Animal Sciences and Veterinary Medicine Cluj-Napoca, 400372 Cluj-Napoca, Romania; alina_vilcan@yahoo.com

* Correspondence: cristina.hegedus@usamvcluj.ro (C.H.); ancuta.rotaru@usamvcluj.ro (A.R.)

**Abstract:** The comparison of the antioxidant activity of the studied seeds of fiber crop (hemp and flax) emphasized a hierarchy of antioxidant capacity. The purpose of the study was to investigate the antioxidant capacity and nutritional value of flax seeds (*Linum usitatissimum* L.) and hemp seeds (*Cannabis sativa* L.) in powder form. In this study, the FT-IR technique was utilized in order to detect molecular components in analyzed samples. Antioxidant capacity was evaluated with photochemical assay as well as humidity, protein, fiber, lipid and carbohydrate content. The FT-IR results reveal the presence of different bio-active compounds in hemp such as flavonoids, tannins, sugars, acids, proanthocyanidins, carotenoids and citric metabolites. The highest antioxidant capacity was observed in flax seeds, 18.32 ± 0.98, in comparison with hemp seeds, 4 ± 0.71 (μg/mg dry weight equivalent ascorbic acid). Regarding nutritional parameters, flax seeds showed the most increased content of protein, displaying average values of 534.08 ± 3.08, as well as 42.20 ± 0.89 of lipids and 27.30 ± 0.89 of fiber (g/100 g/sample). Hemp seeds showed the highest protein content of 33 ± 1.24 (g/100 g/sample).

**Keywords:** antioxidants; flax seed (*Linum usitatissimum* L.); hempseed (*Cannabis sativa* L.); Fourier transform infrared (FT-IR) spectroscopy

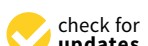

## 1. Introduction

### 1.1. General Aspects about Flax (Linum usitatissimum)

Flax (*L. usitatissimum*) plant has a long history of traditional use both as a source of oil and fiber and is grown for commercial use in over 30 countries of the world. Flax seeds have long been used in human and animal diets and in industry as a source of oil and as the basal component or an additive of various paints or polymers. The beneficial effects are mostly due to flax lipids. Flax oil is the richest plant source of linoleic and linolenic polyunsaturated fatty acids (PUFA), which are essential for humans since they cannot be synthesized in the organism and must be ingested with food [1].

Flax seed (*L. usitatissimum*) proved to be rich in a wide variety of vitamins and minerals (vitamin B1, vitamin B6, folate, calcium, iron, magnesium, phosphorus, potassium) which are extremely beneficial for the body by decreasing the risk of the development of metabolic alterations [2]. This plant is a valuable source of protein, edible oils, α-linolenic acid, dietary fibers [3,4], lignans [5] and secoisolariciresinol diglucoside (SDG) which exhibits a high antioxidant activity [6]. The composition of flax seed of Canadian varieties that dominate

the global production of flax is comprised of: fat—41%, protein—21%, fiber—28%, aromatic acids, lignin and hemicellulose, sugar—6% and ash residue—4% [7]. The uniqueness of *L. usitatissimum* is attributed to a high content of polyunsaturated $\alpha$-linolenic acid (ALA) essential fatty acids in the human diet. The growing interest of researchers and doctors in *L. usitatissimum* is due to the fact that ALA, like hormones, contributes to important biological functions in the human body [8]. It is often used to prevent various diseases, and also used in treatment. In addition, $\alpha$-linolenic acid exhibits vasodilator and anti-inflammatory properties and improves the lipid metabolic profile in humans [9,10].

Hussein et al. found that *flax* oil might be effective in regulating cholesterolemic status and dyslipidemia and has the potential to reduce cardiovascular complications caused by hypercholesterolemia [11]. As a functional food component, flax seed may suppress atherosclerosis due to a reduction in blood circulating cholesterol and was shown to exhibit antiproliferative and anti-inflammatory effects at a cellular level [12,13]. *L. usitatissimum* supplementation decreased total cholesterol, low-density lipoprotein (LDL), apolipoprotein B (ApoB) and apolipoprotein E (ApoE) [13]. Flax seed is also used for animal feed, as a local demulcent and emollient and as a laxative to improve the animal reproductive system and health [14,15].

The health benefits of flax and hemp seeds are related to the high level of polyunsaturated fatty acids (PUFA) present in the seeds. Flax seeds have an unusually high level of alpha-linolenic acid compared to other crops. For this reason, it is expected that flax seed oil can be a substitute for fish oil in the human diet and will find more applications in animal feed and aquaculture [16].

The high level of protein in flax seeds and their high antioxidant capacity have beneficial effects on the digestive tract [17]. A study published in 2009 showed that the introduction of flax seeds into the diet can reduce the concentration of total cholesterol and LDL cholesterol [18].

The high fiber content of flax seed promotes blood pressure and has an effect on intestinal transit problems due to the high content of cellulose and hemicellulose, but may have hypoglycemic and hypolipidemic properties [19–21].

### 1.2. General Aspects about Hemp (Cannabis sativa)

Hemp (*C. sativa*) is one of the oldest plants cultivated in Europe and China (for thousands of years), used for preparing natural remedies, and as a source of food, dietetic oil and cellulose. The content of oil in cannabis sativa seeds amounts to approx. 35%. Hempseed oil is an untraditional oil, which, as a source of essential unsaturated fatty acids and natural antioxidants, may be treated as a product for disease prevention and as a functional food [22].

Another nonconventional food which is being focused on by medicine today is *Cannabis sativa* or hemp. This famous plant is found in the wild flora of Central Asia [23]. Different cannabis varieties with a reduced number of psychoactive cannabinoids and flax seeds are used for the production of fiber and oilseed [24]. *C. sativa* seeds consist of protein (20–25%), carbohydrates (20–30%), fiber (10–15%), oil (25–35%), minerals such as phosphorus, magnesium, potassium, calcium and sulfur, iron and zinc and also a vitamin A precursor carotene [25,26]. Furthermore, the *C. sativa* seed oil structure is mostly based on polyunsaturated fatty acids (<70%). It is considered important among other oils mainly for its optimal linoleic and linolenic acid ratio (3:1), which is considered beneficial for human health. Additionally, with the high concentrations of $\gamma$-linolenic acid (GLA) and steastearidonic acid (SDA), *C. sativa* seed oil is found to be significant throughout the other industrial crops [27,28]. These fatty acids and other components from the oil were found to be efficient against inflammation, ischemic heart diseases, psoriasis, atopic dermatitis and mastalgia, and also regulate biochemical blood parameters, providing stimulation of the general metabolism [26,29–32].

Hemp seeds are high in calories, fat and protein. Hemp seeds contain all 21 fatty amino acids, including those eight amino acids that the human body cannot produce, which

makes hemp a complete protein. These substances are very beneficial for heart health, help the brain develop in children, reduce inflammation in the body, regulate blood pressure, reduce cholesterol and reduce the risk of developing many associated diseases [33].

## 2. Literature Review

The concern of healthy food has always been one of the most important for humanity. Current trends in the composition of a beneficial diet determine the development of new foods with increased biological and physiological value [34,35]. Given the importance of nutrition regarding the appearance of chronic diseases, the recommendations of WHO and medical doctors suggest an increased consumption of plant-derived products is of particular importance and relevance [36,37].

In this context, extraction and evaluation of sandbox tree seed oils were studied by Okolie [38]. Extraction and characterization of vegetable oils from cherry seed by different extraction processes was studied by Straccia [39].

Nyam et al. proposed a study where the physicochemical properties and chemical composition of oil extracted from five varieties of plant seeds (bitter melon, Kalahari melon, kenaf, pumpkin and roselle seeds) was examined by established methods. The thermal properties of extracted oils by differential scanning calorimetry were also evaluated [40].

Essential oils and plant extracts, as well as textile flax and hemp seed, play an important role as raw materials in many products such as cosmetics, food industry constituents and medicines.

*L. usitatissimum* and *C. sativa* are plants with a plethora of biologically active substances. Esculent oils derived from flax, hemp and other plants, depending on the geographical area, are known for their antioxidative medicinal status [41]. Cannabis inflorescences contain cannabinol and therefore in modern traditional medicine, the plant is used as a sedative and anticonvulsant to combat insomnia and epilepsy [42].

Experimental research confirmed the antiepileptic properties of the herb [43–45]. Cannabinoid receptors and their endogenous ligands, known as endocannabinoids, are involved in the regulation of gastrointestinal motility and the regulation of intestinal epithelial integrity, suggesting that cannabis preparations are promising for the treatment of gastrointestinal diseases [46]. Interestingly, biologically active compounds from flax and hemp revealed potential as anticancer drugs after biotechnological analysis [47–49].

Research has been carried out on the development of an ATR FTIR spectroscopic technique (total attenuated reflection with Fourier transform in infrared) for the characterization of cellulosic (vegetable) fibers. Six species of fiber were examined (flax, hemp, jute, cotton, sisal and ramie). Raw fibers were considered, and subsequently, processed fibers from a variety of sources were examined. The technique used allowed us to differentiate the types of fibers based on the relative content of lignin in relation to other cellular components [50].

In their paper, Senilă L. et al. determined the chemical composition in terms of the amount of fatty acids, minerals and proteins and the antioxidant capacity for several food seeds, including flax and hemp [51].

Therefore, the aim of this work was to analyze and obtain comparisons among antioxidant capacity, nutritional value and molecular structures of flax seeds (*L. usitatissimum)* and hemp seeds (*C. sativa*) from the Romanian flora using vibrational spectroscopy techniques (FT-IR), spectrophotometric method and Photochem assay. Fourier transform infrared (FT-IR) spectroscopy is one of the most widely used methods to identify chemical compounds and elucidate chemical structures, and these tools were recently reported for their food and pharmaceutical industry suitability, especially for the description of secondary metabolites and polysaccharide–polyphenol conjugates derived from plant-based extracts [52].

## 3. Materials and Methods

In this research, we analyzed two different varieties of fiber crop (flax seed—*L. usitatissimum* and hempseed—*C. sativa*) according to the analytical information obtained from

dried seeds by means of Fourier transform infrared spectroscopy (FT-IR). The samples were purchased from a local store and stored in dry room prior to further analysis.

### 3.1. Moisture Determination

The determination of moisture content was evaluated by the air-oven method, according to the methodology previously described by Kumar and Balasubrahmanyam, with slight modifications [53]. The weighted sample was introduced into a vial containing lid and conditioned at 105 °C for at least 45 min until a constant mass was obtained and therefore cooled in air-oven. Then, 5–10 g of sample was spread uniformly on the entire surface of the vial and weighted with an accuracy of 0.001 g. The water content was expressed in percentages.

### 3.2. FT-IR Spectroscopy

The seed samples were crushed using a commercial blender, obtaining a powder that was used on the same day.

The sample was prepared using calcined potassium bromide as a matrix material and was mixed at a proportion of 3 mg of the sample (powder of flax seeds and hemp seeds) to 200 mg KBr. Then, the mixture was condensed in 15 mm dies at a pressure equal to 10 t till 2 min. The same procedure was applied for all samples [54].

Measurements were carried out on the infrared scale of 350–4000 cm$^{-1}$ and a spectral resolution was set at 4 cm$^{-1}$ using a Jasco FT-IR-4100 spectrophotometer (Oklahoma City, OK, United States) using the KBr pellet technique. All spectra were acquired over 256 scans. The spectral data were analyzed using Origin 6.0 software (Figure 1). Measurements were carried out on an infrared scale of 500–4000 cm$^{-1}$.

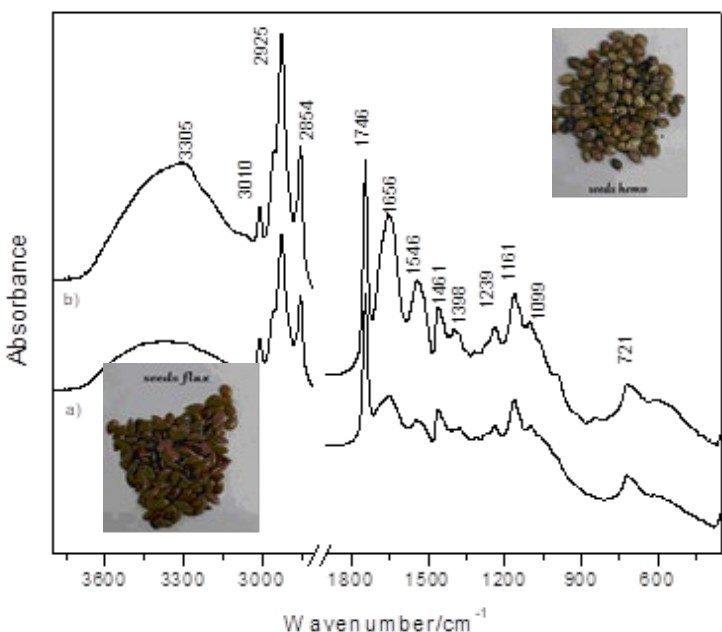

**Figure 1.** FT-IR spectrum of flax seeds (**a**) and hemp seeds (**b**).

### 3.3. Photochemical Assay

The ACL (Antioxidant Capacity of the Liposoluble Compounds) method, previously described by Hegeduş et al., was used to estimate the antioxidant capacity [55]. Photochem$^{®}$ equipment (Analytik Jena AG, Jena, Germany) was used for measuring the antioxidant capacity.

The hydrophilic and lipophilic radical-scavenging activities were measured using the ACL kits, provided by the manufacturer. The extractions for ACL measurements were also carried out according to the manufacturer's protocol using the kits and methanol.

For these measurements, we used 1 g of extract of the sample with 20 mL of methanol, then centrifuged the sample for 5 min (10,000 rpm). The equipment calibration and measurement of samples were based on the inhibition of free radicals. The values of antioxidant capacity were achieved by establishing measurement curves that were compared to the measurement curves obtained for the standard solution. The results are expressed in ascorbic acid equivalents (µg/mg). Data were calculated automatically using a software program called PCL soft.

### 3.4. Determination of Carotenoids

For carotenoid extraction, 1 g of hemp seeds and flax seeds were mixed with an extraction solvent consisting of methanol/ethyl acetate/petroleum ether (1:1:1, *v/v/v*) for exactly 4 h. After filtering the extract, the residue was retracted using the same solvents until the residue became colorless. The extracts were washed with water, diethyl ether and saline solution and the etheric phase was evaporated with a rotary evaporator.

The carotenoid extract was saponified further using KOH 30% in methanol for 6 h. Afterwards, the extract was washed with saline solution, partitioned in a separatory funnel, the etheric phase evaporated and used for further analysis.

Carotenoid quantification reading is the total carotenoid concentration using the spectrophotometric method [56].

### 3.5. Statistical Analysis

IBM SPSS v.19.0 for Windows was used for statistical analysis. Basic statistics were implemented in order to emphasize the arithmetic mean (X) $\pm$ standard deviations (SD) of the content of humidity and antioxidant capacity of the studied forest fruits.

The mean concentration of humidity and antioxidant capacity was compared across the various textile plants using ANOVA, followed by Tukey's test. Differences in the means were considered to be significant when *p*-value < 0.05 [57].

## 4. Results

### 4.1. FT-IR Analysis

The FT-IR spectra were used to identify the functional groups of the macronutrients, including proteins, carbohydrates and lipids in typical spectral regions [58].

Till now, characteristic of flax and onion seeds are the bands obtained from proteins, lipids and carbohydrates, and these are clearly highlighted in the spectra obtained by the FT-IR technique [59].

The band located at 3305 cm$^{-1}$ (Figure 1) corresponds to hemp and flax samples, and seems to be linked to the prediction of the presence of CH stretching of secondary amine.

Figure 1 shows the FT-IR spectrum of flax seeds and hemp seeds, and an interesting region is a characteristic of various types of seeds, it is represented by bands in the range from 2800–3000 cm$^{-1}$. In the case of the spectrum obtained from the hemp, an increase in the intensity of the bands can be observed for 2925 cm$^{-1}$, attributed to antisymmetric stretching of CH, and for 2854 cm$^{-1}$, attributed to symmetric stretching vibrations of CH, compared to that obtained from seeds of linen [60].

Both spectra for flax and hemp showed the typical carbonyl band at 1746 cm$^{-1}$ and those of hydrogen/carbon bond (alkene, alkane) stretch in the region 2700–3010 cm$^{-1}$, which were obviously dominant in the hemp sample and progressively less intense in the flax sample. The spectra seeds presented a band typical of the protein amide I due to C=O stretching vibration at 1656 cm$^{-1}$, and this band is more evident in the spectrum obtained from hemp [37].

Comparing the spectra obtained from the two types of seeds, a high-intensity band can be observed at 1546 cm$^{-1}$, vibrationally attributed to the amides [61].

The 1161 cm$^{-1}$ band attributed to vibrational C-O-C asymmetrical stretching is due to the presence of cellulose/hemicellulose in hemp seeds and is more intense compared to the band obtained from the flax seed spectrum [62].

Carbohydrates possess characteristic IR absorptions between 1200 and 750 cm$^{-1}$ relevant to coupling and the combination of stretching and deformation or vibrational modes of individual bonds in the molecular skeleton. The scissoring of C-H showed and a band was identified at 1461 cm$^{-1}$, the stretching of C-O corresponding to 1239 cm$^{-1}$ and 1161 cm$^{-1}$ with medium intensity. The absorption bands for rocking of CH$_2$ may be found at 721 cm$^{-1}$ [63].

### 4.2. The Nutritional Quality of the Studied

The nutritional values for 100 g of two types of textiles plants and the characteristic differences among them are presented in Table 1.

**Table 1.** Nutritional parameters of the investigated seeds (100 g/sample).

| Samples | Calories | Lipids | Protein | Fiber |
|---|---|---|---|---|
| Flax (*L. usitatissimum*) | 534.08 ± 3.08 kcal | 42.20 ± 0.89 g | 18.30 ± 0.73 g | 27.30 ± 0.89 g |
| Hemp (*C. sativa*) | 514.12 ± 2.05 kcal | 41.02 ± 1.39 g | 33.04 ± 1.24 g | 4.07 ± 0.48 g |

In flax seeds was identified the highest concentration of lipid fraction and calories. The flax seeds had the highest carbohydrate and fiber content in comparison with their content in hemp. In addition, hemp showed a higher content of protein in comparison to flax.

Carotenoids were identified in hemp seeds powder and the total carotenoid concentration from hemp was calculated according to the formula:

$$X \text{ (mg carotenoids)} = (A \times V \times 100)/(2500 \times 1 \times 100)$$

where:

A = the absorbance at 450 nm;
V = sample volume (mL);
2500 = molar absorption coefficient (E1%);
l = 1 cm.
Following the calculations, we obtained X = 3.02 μg/g
For flax seeds, the amount of carotenoids was calculated with the same formula as in the case of hemp seeds, and following the calculation, we obtained X = 4.65 μg/g.

These differences reported in the nutritional parameters of the studied textile plants are the consequence of several reasons, the most relevant being the species, the variety and pedo-climatic conditions of culture.

### 4.3. Statistical Analysis for the Water Content and Antioxidant Capacity

ANOVA analysis was followed by Tukey's test in order to perform multiple comparisons regarding the antioxidant capacity of the considered forest fruits.

The Tukey test revealed significant differences between the mean amount of antioxidants contained, respectively, in the two categories of textile plants considered. The results of Tukey's test are displayed in Table 2.

**Table 2.** The content of water and antioxidant capacity in tested seeds.

| Samples | Water Content (%) | Antioxidant Capacity (μg/mg Equivalent Ascorbic Acid) |
|---|---|---|
| Flax (*L. usitatissimum*) | 0.78 ± 0.09 | 18.32 ± 0.98 |
| Hemp (*C. sativa*) | 1.81 ± 4.95 | 4 ± 0.71 |

The significance was determined at $p < 0.05$. The results are expressed or plotted as the mean values ± standard deviation. The test revealed significant differences between the mean amounts of calories, lipids and protein contained, respectively, in *L. usitatissimum* and *C. sativa* (Table 1). Additionally, the amount of fiber contained in *L. usitatissimum* was significantly higher than that reported in *C. sativa*.

Relationships between the antioxidant capacity and water content of *flax* and *hemp seeds* are presented in Table 3. We obtained a strongly significant positive correlation at the $p < 0.05$ level between the antioxidant capacity and water content of considered seeds.

**Table 3.** Linear correlation coefficients between antioxidant capacity and water content of tested seeds.

| Samples | Antioxidant Capacity (µg/mg Equivalent Ascorbic Acid)—*L. usitatissimum* | Antioxidant Capacity (µg/mg Equivalent Ascorbic Acid)—*C. sativa* |
|---|---|---|
| Humidity—Flax (*L. usitatissimum*)% | 0.92 * | - |
| Humidity—Hemp (*C. sativa*)% | - | 0.83 * |

Note: * Correlation is significant at $p < 0.05$.

## 5. Conclusions

The data reveal that the flax samples have the highest antioxidant capacity when compared to hemp samples. Flax samples from Romanian flora exhibit an antioxidant capacity and more investigations should be performed in order to describe the complex interactions between antioxidants and the human body. It is promising to observe that molecular compounds, such as those in hemp and flax, may be of potential interest in food and pharmaceutical research.

The comparison of the antioxidant activity of the studied textile plants emphasized a hierarchy of antioxidant capacity in flax seeds and hemp seeds.

The resulting data reveal that the hemp and flax seeds have the highest antioxidant capacity. We obtained a strong significant positive correlation at the $p < 0.05$ level between the antioxidant capacity and humidity content of considered seeds.

With the help of FT-IR spectrometry, it was possible to make physicochemical determinations on the analyzed samples.

According to the FT-IR spectra obtained, it could be highlighted that hemp seeds have a higher protein intake ($1546 \text{ cm}^{-1}$), the results being in accordance with Table 1.

Additionally, following FT-IR spectroscopy, it was observed that hemp seeds have a higher concentration of cellulose/hemicellulose compared to flax seeds.

This paper, which compares the two types of seeds by FT-IR spectroscopy, emphasizes once again the benefits of these plants on the health of the human body and also in the animal field.

Flax seeds contain fiber, protein, amino acids, Omega-3 fatty acids, vitamins and minerals such as vitamin B1, copper, magnesium and phosphorus, and also other compounds essential for the proper functioning of the body. In nutrition, they are considered a very beneficial food for the body.

Constant consumption of flax seed plays an important role in regulating blood cholesterol levels.

Substances found in flax seeds, such as Omega-3 fatty acids, fiber and lignans, help maintain heart health by reducing blood pressure and relieving arthritis symptoms due to their anti-inflammatory properties.

Additionally, due to their insoluble fiber content, they have an effect on blood sugar levels and a beneficial effect on regulating intestinal transit.

Hemp seeds contain protein, fiber and healthy fatty acids with antioxidant effects. All the nutrients in the composition of these seeds help maintain the health of the heart, skin and digestive system.

In their composition are found essential fatty acids, such as alpha-linolenic (Omega-3 fatty acid) and gamma-linolenic (Omega-6 fatty acid). The proportion of the two fatty acids is very important for the body, and hemp seeds provide these acids in a balanced way.

Hemp seeds also contain fiber, soluble and insoluble, which has many roles in the body: reduce appetite, regulate intestinal transit, stabilize blood sugar and support the health of the intestines.

In terms of minerals and vitamins, hemp seeds are a good source of magnesium, phosphorus, potassium, iron, zinc, vitamin E and B-complex vitamins (niacin, riboflavin, thiamine, pyridoxine and folic acid).

**Author Contributions:** Conceptualization, L.A., A.R. and C.H.; methodology, L.A.; software, L.A., A.R. and R.S.; validation, I.D.P., Z.D. and L.A.; formal analysis, L.A.; investigation, Z.D.; resources, L.A.; data curation, A.R. and C.H.; writing—original draft preparation, L.A., A.R. and C.H.; writing— review and editing, R.S. and C.H.; visualization Z.D., A.T. and R.S.; supervision, I.D.P. and L.A. The first author, L.A., and the corresponding authors, A.R. and C.H., have equal contribution to this scientific paper, all three being considered as first author. All authors have read and agreed to the published version of the manuscript.

**Funding:** The publication fee was supported by university funds.

**Informed Consent Statement:** Not applicable.

**Data Availability Statement:** The data presented in this study are available in the article.

**Conflicts of Interest:** The authors declare no conflict of interest.

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
