# Peer review of "Characterization of Flax and Hemp Using Spectrometric Methods"

_applsci, doi:10.3390/app11188341_

Round 1

Reviewer 1 Report

Review: Andronie et al. 2021

Submitted to Applied Sciences

            This manuscript describes the characterization of local hemp and flax seed by FT-IR and biochemical methods to determine seed calorie, lipid, protein, fiber, moisture, and antioxidant content. Primarily descriptive in nature, the results and conclusions of this work do not present a compelling argument for publication in its current form. The differences between hemp and flax seed should be more strongly presented, and the significance of the results for agriculture and health should be much better defined.

A number of comments:

  • Only when a taxonomic name is first presented should it be written out in its entirety (e.g. Linium usitatissimum). Following this first instance the genus name should be abbreviated (e.g. L. usitatissimum).
  • The treatment of significant digits in the data is haphazard. For example, the value of 534 ± 3.08 (g/100 g/sample) would suggest that the data are precise to five significant digits. Wonderful if this is the case, but one should ensure that the precision of the reported data reflects the precision of the methods and instruments used to collect the data.
  • In the Introduction, flax and hemp seed should be given their own, distinct sections (something like ‘1.1 Flax ’ and ‘1.2 Hemp’).
  • Given the enormous variety of compounds in seed and the differences in growth conditions of the two plant species, this reviewer is skeptical of how quantitative the FT-IR method is for comparing seed contents. The Introduction indicates that the molecular components of the seeds would be presented, but very little detail is then given in the Results, mostly referring to the atomic bonds detected. The possible seed sources of FT-IR spectra differences should be discussed in more detail. Also, is there any significance to the peak difference at 1546 cm-1?
  • In the spectrophotometric determination of seed carotenoid content, are there any interfering species that would also absorb at 450 nm? If so, how were these accounted for?
  • Seed ‘humdity’ is typically referred to as ‘water content’. Also, there are no units for this presented in Table 2.
  • It is unclear how the linear correlation coefficients in Table 3 were obtained. This reviewer cannot determine how they were derived from the results of Table 2.
  • The Conclusions section is quite short, and only discusses seed antioxidant activity. For a stronger manuscript this needs to be expanded and improved.

Author Response

The manuscript was rewritten, in the introductory part two sections were made, one for linen and one for hemp. New references from the specialized literature were added, the discussion and conclusions part was completed.

Reviewer 2 Report

The authors investigated the antioxidant capacity and nutritional value of flaxseeds and hemp seeds in powder form. The FT-IR technique was utilized in this study to detect molecular components in analyzed samples. Antioxidant capacity was evaluated also with photochemical assay as well as humidity, protein, fiber, lipid, and carbohydrate content. The methodology and experiments are consistent. English grammar and spelling need improvement. The discussion part should be added and I suggest the authors clarify the differential of their work compared to the related work in this suggested part.

Author Response

(The authors gave the same response as above.)

Reviewer 3 Report

The manuscript “applsci-1350546” deals with the characterization of hemp and flax using spectrometric methods. The topic is interesting and relevant, and after minor revision, I recommend the publication of this work in Applied Science.

General Comments:

According to the authors, the aim of the work was to analyze and obtain comparisons among anti-oxidant capacity, nutritional value and molecular structures of flaxseeds (Linum usitatissimum) and hempseeds (Cannabis sativa) from the Romanian flora using vibrational spectroscopy techniques (FT-IR) spectrophotometric method and photochem assay. The Applied Science journal is an open access journal on all aspects of applied natural sciences. Therefore, the there are some important issues, which need to be addressed:

1) The authors described in the manuscript title and in the text the use of spectrometric methods. However, I only found the use of FT-IR method on manuscript. Therefore, the authors should rewrite this information.

2) It is a well-developed study, well-written version, and with relative scientific quality. However, the authors should improve the study novelty and benefits in the text (make them clear), since there are others studies covering the respective subjective.

3) For the authors impove the manuscript quality, multivariate statistical analysis should be developed in order to achieve a better correlation among the results.

Author Response

(The authors gave the same response as above.)
